# Chemiresistive Properties of Imprinted Fluorinated Graphene Films

**DOI:** 10.3390/ma13163538

**Published:** 2020-08-11

**Authors:** Vitalii I. Sysoev, Mikhail O. Bulavskiy, Dmitry V. Pinakov, Galina N. Chekhova, Igor P. Asanov, Pavel N. Gevko, Lyubov G. Bulusheva, Alexander V. Okotrub

**Affiliations:** 1Nikolaev Institute of Inorganic Chemistry SB RAS, 3 Acad. Lavrentiev Ave., 630090 Novosibirsk, Russia; mikhail@bulavsky.pp.ru (M.O.B.); pinakov@niic.nsc.ru (D.V.P.); chekhova@niic.nsc.ru (G.N.C.); asan@niic.nsc.ru (I.P.A.); paul@niic.nsc.ru (P.N.G.); bul@niic.nsc.ru (L.G.B.); 2Faculty of Natural Sciences, Novosibirsk State University, 2 Pirogova Str., 630090 Novosibirsk, Russia

**Keywords:** fluorinated graphene, resistivity, nitrogen dioxide, adsorption, gas sensor

## Abstract

The electrical conductivity of graphene materials is strongly sensitive to the surface adsorbates, which makes them an excellent platform for the development of gas sensor devices. Functionalization of the surface of graphene opens up the possibility of adjusting the sensor to a target molecule. Here, we investigated the sensor properties of fluorinated graphene films towards exposure to low concentrations of nitrogen dioxide NO_2_. The films were produced by liquid-phase exfoliation of fluorinated graphite samples with a composition of CF_0.08_, CF_0.23_, and CF_0.33._ Fluorination of graphite using a BrF_3_/Br_2_ mixture at room temperature resulted in the covalent attachment of fluorine to basal carbon atoms, which was confirmed by X-ray photoelectron and Raman spectroscopies. Depending on the fluorination degree, the graphite powders had a different dispersion ability in toluene, which affected an average lateral size and thickness of the flakes. The films obtained from fluorinated graphite CF_0.33_ showed the highest relative response ca. 43% towards 100 ppm NO_2_ and the best recovery ca. 37% at room temperature.

## 1. Introduction

Graphene materials are widely examined as sensitive layers of the resistive gas sensors owing to a strong dependence of the electrical properties on the surrounding medium [1]. It has been demonstrated that graphene-based materials are able to detect various electron-donor (NH_3_ [2], H_2_S [3], CO_2_ [4]) and electron-acceptor (NO, NO_2_ [5,6], organic nitro-compounds [7]) gases or vapors. Most of the studies are conducted with nitrogen dioxide NO_2_, as it is a convenient model system. In addition, detection of nitrogen oxides gases is significant in the protection of environmental and human health and the detection of explosives [8,9]. The main disadvantages of graphene-based sensors are slow response and poor recovery that makes them useless for real time sensing at room temperature. The reasons for long response and recovery time are slow adsorption kinetics of molecules and presence of high-energy sites, such as defects and oxygen containing groups [10], which catch adsorbate thus making the sensor recovery difficult. On the other hand, structural defects, edge states, dopants, and functional groups play an important role in the electrical response of graphene [6,11,12,13,14]. The controlled introduction of a certain type of defect and functional group enhances the performance of a graphene-based sensor by increasing the binding energy of adsorbate and charge transfer in the reactive sites. Functionalization and doping of graphene include noncovalent modification via π–π stacking, covalent attachment to the unsaturated double bonds, intercalation [15], and insertion of foreign elements in the graphene lattice [16]. The direct grow of functionalized or doped material is a challenging task, and it is limited by precursor selection. The post-synthesis functionalization of graphene prepared using the chemical vapor deposition (CVD) method always requires involvement of complicated techniques, such as ion bombardment, plasma treatment, etc. The use of graphite and graphite derivatives is a more reasonable route for preparation of functionalized graphene.

Halogenation is a promising way to obtained graphene derivatives possessing an energy gap, unique two-dimensional structure, and uniform distribution of functional groups on the surface [17]. As compared to bulky chlorine [18] and bromine [19] atoms, fluorine can attach to every carbon atom up to a fully fluorinated graphene (FG) layer [20]. A strong covalent bonding between carbon and fluorine distorts an initially planar graphene sheet [21] thus creating the active sites for the adsorbate. The amount, distribution, and character of C–F bonds strongly affect the chemical, electrochemical, electrical, electronic, optical, and magnetic properties as well as stability and hydrophobicity of FG [22]. There are different ways of fluorination, such as a treatment in hydrofluoric acid solution [23], hydrothermal methods [24], photonic [25], plasma [26], and by using the gaseous fluorinating agents [22,27]. These methods differ by the efficiency, resultant fluorine pattern [28], and structure of obtained graphite or graphene derivatives. Moreover, single- or few-layered FG can be obtained using liquid phase exfoliation of fluorinated graphite in an organic solvent. The deep understanding of the CF_x_ structure is fundamental to control the physicochemical properties of these materials by changing the number of layers, size, and surface chemistry, which determine an application field of the fluorinated graphene.

Herein, we report the preparation of FG films from fluorinated graphites with a composition of CF_0.08_, CF_0.23,_ and CF_0.33_ using a sonication of powders in toluene followed by a vacuum filtration. The films deposited on membranes can be transferred on various substrates, including a flexible polymer substrate by simple imprinted technique. The structure and composition of the FG films were studied by scanning electron microscopy (SEM), energy dispersive X-ray spectroscopy (EDS), atomic force microscopy (AFM), and Raman scattering. We showed that both the fluorination degree and the particle size affect the sensor properties of the films toward gaseous NO_2_. The functionalization modifies the electronic state of graphene, improving response and recovery of the sensor. Better exfoliation ability of highly fluorinated graphite provides a greater increase in the sensor response as compared to the lower fluorinated films.

## 2. Materials and Methods

Natural graphite, purified from impurities, was fluorinated in Teflon reactors using vapors of BrF_3_ and Br_2_ at room temperature. The fluorine content in the product was tuned by changing the concentration (C) of BrF_3_ diluted by Br_2_ and the reaction time (t). The details of the synthesis procedure are described elsewhere [29,30]. Three samples with different fluorine content were used in this study. The composition of the fluorinated layers determined from elemental analysis was a CF_0.33_ (C(BrF_3_) = 10 wt.%, t = 30 days), CF_0.23_ (C(BrF_3_) = 4.3 wt.%, t = 90 days), and CF_0.08_ (C(BrF_3_) = 2.1%, t = 150 days. Electronic state of the elements in the samples was revealed by X-ray photoelectron spectroscopy (XPS) on a Phoibos 150 Specs spectrometer using monochromatic Al Kα (1486.6 eV) radiation. The surface composition was determined from the survey XPS spectra; the high-resolution spectra were fitted using Doniac-Sunjic and Gaussian-Lorentzian peak profiles after subtraction of Shirley-type background.

A fluorinated graphite powder (ca. 1 mg) mixed with toluene (10 μL) was grinded in an agate mortar and then bath-sonicated in toluene (10 mL) for 30 min. Non-exfoliated particles were removed by sedimentation of dispersion for 1 h. 9 mL of the obtained stable dispersion of FG was filtered under vacuum through a nitrate cellulose (CN) membrane with a diameter of 47 mm and a pore size of 0.45 μm. The obtained films were designated as FG-s. To increase the degree of exfoliation of fluorinated graphite, the grinded powder was sonicated in 40 mL of toluene for 30 min. After sedimentation for 1 h, the dispersion was sonicated at 78 W for 1 h and centrifuged at 1000 rpm for 15 min to separate the fine particles of FG. The supernatant was additionally sonicated at 78 W for 15 min and 35 mL of the dispersion was filtered to obtain the films denoted FG-c.

FG-s and FG-c films were imprinted to a SiO_2_/Si, glass or polyethylene terephthalate (PET) substrate using hot pressing. The CN membrane with a deposited FG film was cut into pieces of 3 × 5 mm^2^, and a piece was put on a substrate with a size of 6 × 5 mm^2^ by the side with the film. The substrate was heated to 120 °C, and the load of 1 kg/cm^2^ was applied to the assembly for 15 min. The CN membrane was mechanically detached from the FG film, and membrane residues were removed by acetone. To increase the adhesion with silicon or glass substrate, the films were wetted by distilled water. Finally, electrodes were deposited by silver conductive paint (RS Components Ltd, Corby, UK) on two opposite sides of the film that gave a sensitive layer with a size of ca. 3 × 3 mm^2^. The samples were annealed at 80 °C, as long as the conductivity did not stop changing.

The sensor test was performed in the experimental gas system described in our previous work [31]. Electrical characterization of the films was carried out using a two-probe technique under atmospheric pressure. The current was recorded using a Keithley 6485 picoammeter at a constant voltage in the range from 0.1 to 1 V, depending on film conductivity. The standard cycle of the sensor test consisted of 3 min of exposure to gas mixture (100 ppm NO_2_ in Ar) and 12 min of recovery in pure Ar. The relative response and recovery were estimated using Equations (1) and (2):(1)$$Relative\;response\;\left(\%\right)=\frac{I_{g}-I_{0}}{I_{0}}\times 100\%$$
(2)$$Recovery\;\left(\%\right)=\frac{I_{g}-I_{a}}{I_{g}-I_{0}}\times 100\%$$
where *I_o_* and *I_g_* are the currents before and after exposure to NO_2_ and *I_a_* is the current after sensor recovery by pure argon.

The characterization was made for FG films on silicon substrates. Raman spectra were recorded using an excitation from an Ar^+^ laser at 514 nm on a LabRAM HR Evolution (Horiba, Kyoto, Japan) spectrometer. Optical absorbance measurements were performed using an Optizen 220 UV spectrometer (KLAB, Daejeon, Korea). Elemental analysis of the FG films was carried out by EDS on a Bruker QUANTAX spectrometer with an XFlash 6|60 detector. Morphology, thickness, and particle sizes of FG films were characterized using a set of microscopic methods, namely optical microscopy on a BX 51TRF microscope (Olympus Corporation, Tokyo, Japan), SEM on a JEOL JSM-6700F microscope (Tokyo, Japan), and AFM on a Solver Pro (NT-MDT) microscope (Moscow, Russia). The AFM measurements were performed in tapping mode using cantilevers NSG10 (NT-MDT) with a tip curvature radius of 6 nm and an average value of the force constant of 11.8 N/m.

## 3. Results

### 3.1. Materials Characterization

The content of fluorine in the surface of the fluorinated graphites determined from the XPS survey spectra was ca. 7 at.% in CF_0.08_ sample, ca. 19 at.% in CF_0.23_ sample, and ca. 25 at.% in CF_0.33_ sample. Lower values given by XPS are due to partial hydrolysis of the sample surface by H_2_O present in laboratory air [32].

The XPS C 1s spectra of the samples are compared in Figure 1a. Each spectrum exhibits a set of the components corresponding to (1) sp^2^-hybridized carbon (ca. 284.5 eV), (2) carbon atoms located at C–F bonds (C*–C(F)) (ca. 285.3 eV), (3) sp^3^ carbon atoms covalently bonded with fluorine atoms (C*–F) (ca. 288.0 eV), and (4–6)–CF_x_ (x = 1–3) groups (ca. 289.7–292.0 eV) located at crystallite boundaries [33,34,35,36]. A broad high-energy component is assigned to π→π* electron excitations. The contribution of Csp^2^ component to the C 1s spectrum decreases in the set CF_0.08_ > CF_0.23_ > CF_0.33_, and this indicates a reduction of the average size of graphene-like areas remaining in the layers after the fluorination procedure. A comparable integral intensity of the components C*–F and C*–C(F) means the formation of chains from fluorine atoms [37]. The co-existence of patterns from alternating CF chains and bare carbon chains, and graphene islands in the fluorinated layers is typical for the sample obtained by room temperature fluorination of graphite [33].

The XPS F 1s spectra of all the studied fluorinated graphites exhibited a dominant peak at ca. 687.4 eV (Figure 1b), attributing to weakened covalent C–F bonding [38]. A broad component around ca. 693.0 eV corresponds to the loss-energy satellite. The components between the main peak and the satellite are assigned to –CF_x_ groups located at the edges of graphite crystallites [35]. The fraction of these groups is largest (~40%) in the CF_0.23_ sample, as both F 1s and C 1s spectra are revealed. It is likely that the holding of graphite in the vapors of highly reactive BrF_3_ for several months causes etching of graphene layers at defects and grain boundaries accompanying by the formation of –CF_x_ groups. Stronger etching observed for CF_0.23_ can be related to the higher concentration of BrF_3_ (4.3 wt.%) in the reaction mixture as compared to that used for the synthesis of CF_0.08_.

The method used here for the fluorination of graphite leads to the covalent attachment of fluorine atoms to both sides of the graphene planes [32]. As a result, the interlayer distance in fluorinated graphites increases, while the van der Waals forces between the layers decrease as compared to graphite. Weakening of the interaction between the layers allows easy exfoliation of fluorinated graphites in liquid media [39,40]. We tried tetrahydrofuran, chloroform, acetonitrile, isopropanol, and toluene. Freshly prepared dispersions of the fluorinated graphites have a yellowish color. This color changed under sonication of the dispersions in acetonitrile and tetrahydrofuran due to partial reduction of fluorinated graphene layers. An ultrasound treatment of the samples in toluene caused the most effective splitting of the layers among the rest of the solvents. Figure 2a shows colored dispersions of CF_0.08_, CF_0.23_, and CF_0.33_ in toluene, which were stable after about an hour. The dispersion has a darker color with the decrease in fluorine content in parent fluorinated graphite. Optical absorption spectra measured for the dispersions confirmed the reduction of light transmittance for the low-fluorinated samples (Figure 2b).

Large agglomerates precipitate within an hour of sedimentation of the dispersion, and their removal produces supernatant, which is stable for several hours. To increase the fraction of few-layered FG, additional stages of sonication and centrifugation were used. Optical microscopy showed that this treatment decreases the average size of the FG flakes from 3–20 to 1–5 μm (Figure 3a,b). Moreover, the flakes become more transparent, which undergoes a decrease in the number of adjacent layers. Filtration of FG dispersions after sedimentation of large aggregates and those after additional centrifugation step yielded the films on the CN membranes denoted FG-s and FG-c, respectively. These films can be transferred to various substrates. SEM images of the films deposited from CF_0.08_ and CF_0.23_ parents on silicon substrates are compared in Figure 3c,d, respectively. Both films consist of wrinkled flakes, however the flakes in less fluorinated material have a flatter structure. Covalent C–F bonds destroy the conjugated π-system of graphene planes, thus reducing their rigidity. Soft layers fold under mechanical treatment in solvent, and the degree of this deformation depends on the number and distribution of C–F bonds. Transparency of an FG film deposited on the PET substrate demonstrates an advantage of the proposed approach to obtain thin graphene-based layers (Figure 3e). The black stripes are electrodes painted by silver paste. Integrity of the film is maintained when the substrate is bent. The thickness of the films obtained was ~150–250 nm as determined by AFM (Figure 3f).

Raman spectroscopy proved the functionalization of graphene layers in studied materials (Figure 4). Fluorination of graphite led to the appearance of defect-activated peaks D at 1351 cm^−1^ and D’ at 1604 cm^−1^ (Figure 4a), which correspond to the single-phonon intervalley and intravalley scattering processes, respectively. The intensity of these peaks relative to the G peak intensity changes non-monotonically with the fluorine content, which is similar to the previous results on functionalized graphenes [41]. The D’ peak is separated from G peak in the spectrum of CF_0.08_ powder, and these peaks are merged when the fluorine content increases. All the single-phonon peaks are broadened for the FG films (Figure 4b) as compared to the powder parent materials. The intensity of D peak increases significantly due to the formation of new edge states resulting from the exfoliation process. An increase in the fluorination degree leads to a blue shift of the two-phonon G’-peak for both powders and FG films, which evidences p-type doping of graphene [42].

### 3.2. Sensor Tests

Effect of the size of FG particles on the sensing performance was examined on the example of FG-s and FG-c films obtained from CF_0.23_ sample. The tests were performed toward 100 ppm of NO_2_ in argon. Figure 5a compares dynamic change of the relative current of the films during five cycles of exposure to NO_2_, followed by purging with pure argon. The FG-s sensor showed ca. 23% increase in the conductivity in the first cycle. The FG-c film obtained from a smaller fraction of the particles had approximately 1.5-times higher change in the conductivity. The relative response and recovery of the films estimated using Equations (1) and (2) are presented in Figure 5b,c, respectively. Both samples showed a gradual decrease in the response with the cycling (Figure 5b), which could be attributed to the nonreversible adsorption of NO_2_ at room temperature. The recovery behavior of the sensors was opposite to the response behavior. The FG-s sample showed recovery ca. 30%, which was higher than that for FG-c (Figure 5c). Similar dependences of the relative response and recovery on the particle size were also observed for the FG sensors obtained from CF_0.08_ and CF_0.33_ samples.

The sensor performance, depending on the fluorine loading, was studied for the thinner FG films obtained after the centrifugation step. Figure 5d compares the run-to-run tests of the FG-c films obtained from the fluorinated graphites with different fluorine content. The largest changes in the conductivity under the adsorption of NO_2_ molecules were detected for the FG-c film obtained from CF_0.33_ sample. The relative responses of the FG-c sensors in five consecutive operation cycles are presented in Figure 5e. In the first cycle, the response was 21, 28, and 42% for the sensors prepared from CF_0.08_, CF_0.23_, and CF_0.33_ samples, respectively. Then, the values decreased, whereas the dependence on the fluorination loading remained. Moreover, the reversibility of the signal at room temperature was strongly improved with the increase in fluorine content in FG film due to better recovery of the sensor (Figure 5f).

The FG-c films obtained from the fluorinated graphites with the lowest and highest fluorination degree, CF_0.08_ and CF_0.33_, were additionally tested at an operation temperature varied from 30 to 80 °C with a step of 10 °C. At each temperature, one standard cycle was performed to determine the response of the sensor to 100 ppm of NO_2_ in argon. After the measurement, the sample was heated to 80 °C and annealed at this temperature for 1 h to achieve the desorption of NO_2_, and then it was cooled down to the required operation temperature. Both sensors showed a faster response and recovery with the rise of the operation temperature. The response gradually decreased from 23 to 8.9% for the FG-c sensor prepared from graphite fluoride CF_0.08_ and from 33 to 20% for the FG-c sensor from CF_0.33_ when the temperature changed from 30 to 80 °C (Figure 6). The higher fluorinated sample showed much better recovery at all temperatures, which reached a maximum level of ca. 80% at 80 °C after 12 min of sensor purging with pure argon.

## 4. Discussion

Since Novoselov and coworkers [1] reported the detection of a single NO_2_ molecule on mechanically exfoliated graphene, sensor properties of graphene materials as an active material are widely investigated. It was shown that graphene exhibits maximum sensitivity to charge carriers corresponded to neutrality point. On the other hand, ideal graphene is chemically inert and does not interact specifically with molecules [43]. The fluctuations in the spectral density of the low-frequency current induced by some gases were proposed as the sensing parameter to enhance selectivity of graphene [44]. Surface functionalization and operation at elevated temperatures reduce the response and recovery times of graphene-based sensors [45]. These sensing parameters can be controlled by the density of the adsorption sites [46]. Fluorination of graphene is especially suitable in that case, because it introduces single-type functional groups at the basal plane and allows tuning the F/C ratio. The electrical transport properties of partially fluorinated graphene can be adjusted by tuning the fluorination degree for application in chemical sensors. The charge carrier mobility for single flake devices was found to increase with an increase in the fluorination degree, reaching 2000–3000 cm^2^V^−1^s^−1^ [47]. In case of thin films, the charge transport is largely affected by edge/edge, edge/plane, and plane/plane junctions. Additionally, we have previously demonstrated that C–F groups modify the surface chemistry of graphene forming specific sites for molecule adsorption [31,48]. A similar effect has been achieved for plasma-fluorinated CVD-graphene, whose enhanced sensor performance at room temperature was attributed to p-type doped nature of FG and stronger physical adsorption of ammonia [26]. Park and co-authors have proposed to modify graphene oxide by fluorine and achieved better sensitivity of the fluorinated sensor to ammonia, however all obtained sensors failed to be recovered at room temperature [49].

Covalent attachment of fluorine to the basal plane and the edges of graphene turns part of the carbon atoms from sp^2^- to sp^3^-hybridization state. The XPS study of graphite fluorides used in the present work for the preparation of FG films showed a decrease in the fraction of sp^2^-hybridized carbon regions and an increase in the population of sp^3^ defects with the synthesis time and concentration of BrF_3_ in the reaction mixture (Figure 1). These defects led to reliable p-type doping of the graphene layers and, as a result, to an increase in the resistivity of FG films (Table 1). Such behavior is consistent with previous works on the fluorination [47] and oxidation [50] of graphene. For the FG films obtained using the sedimentation procedure without or with the centrifugation treatment, we observed a decrease in the conductivity by more than three orders of magnitude when the fluorine content in parent fluorinated graphite increased from ca. 7 to ca. 25 at%. The films obtained from the centrifuged dispersions of FG samples showed a lower resistivity for all fluorination degrees due to partial fluorine detachment with an increase in the sonication time. The EDS determined that fluorine content changes from ca. 19 at.% for graphite powder CF_0.23_ to ca. 15 at% for the FG-s film and ca. 14 at% for the FG-c film.

Exposure of the FG films to the electron-acceptor NO_2_ molecule resulted in an increase in the conductivity, which undergoes an increase in the number of charge carriers. We examined the influence of particle size on the sensor response of FG films, obtained from sedimented and centrifuged dispersions of graphite fluoride CF_0.23_. Run-to-run cycling showed that the small-size fraction has a higher sensor response compared to the large-size fraction (Figure 5b). This could be attributed to more effective influence of the adsorbed molecule on the charge states of graphene particles due to higher surface–volume ratio [51]. It was shown that monolayer and bilayer graphene have the highest sensitivity to adsorbed molecules [11,52], and the same tendency was observed for carbon nanotubes, whereas the sensitivity dropped down from single-walled to multi-walled nanotubes [53]. The recovery of the FG sensors (Figure 5c) showed behavior opposite to the response. As compared to the FG-s sensor, the centrifuged particles had a lower degree of recovery. Probably, small particles formed a more developed pore structure, which trapped the NO_2_ molecules.

Response and recovery of the FG sensor increased with the fluorine content (Table 1). Fluorinated graphene is a p-type semiconductor, since fluorine is strong electron acceptor [22]. An increase in the content of covalently attached fluorine leads to an increase in the concentration of hole carriers in the material. Additionally, fluorination introduces adsorption sites near the fluorine groups [26,48]. To further highlight the influence of fluorine attachment to the graphene plane on the sensor performance of FG films, we carried out experiments at temperatures between 30 and 80 °C (Figure 6). An increase in operation temperature resulted in faster saturation of response, which was achieved within ca. 3 min at 80 °C for the FG-c sensor prepared from CF_0.33_. Additionally, we observed retention of the response with an increase in the operation temperature for the sensors. The observed trend could be related to two factors. The first factor is an increase in the concentration of charge carriers in semiconducting materials with temperature [50]. The second factor is the change in conductivity due to an increase in the adsorption rate of molecules on the surface of FG films. We observed the higher impact of the first factor for both sensors, resulting in a decrease in the response ca. 1.5 and ca. three times for the FG-c sensor based on CF_0.33_ and CF_0.08_, respectively, with an increase in the operation temperature from 30 to 80 °C. The lower response retention for the former sensor undergoes a higher adsorption rate of NO_2_ molecules (Figure 6). The higher fluorination degree in that sensor also resulted in better recovery level of 80% for 12 min at 80 °C, which is comparable with the recovery of reduced graphene oxide at 150 °C [54]. This difference could be attributed to lower adsorption energy of NO_2_ on the fluorinated graphene. Other two-dimensional materials, particularly MoS_2_, WS_2_, and black phosphorus, are widely examined for gas sensing due to their high surface–volume ratio and tunable band gap [55]. Unlike the FG, the fabrication of sensors from the above materials is currently costly. Contaminations and easy oxidation of surfaces require the capping step to prevent problems with reliability and stability of these devices [56].

## 5. Conclusions

Fluorination by a BrF_3_/Br_2_ gaseous mixture at room temperature was used to synthesize fluorinated graphite with a tunable F/C ratio by changing the synthesis time and concentration of BrF_3_. Fluorinated graphites are easily exfoliated in toluene forming stable suspensions when the fluorine content was higher than 7%. The simple imprinting method was developed to prepare uniform FG films with a thickness of ~200 nm on different substrates. Thin FG films can be transferred on flexible polymer substrates to obtain transparent and bendable films. We revealed the influence of particle fraction and functional composition of FG on the performance of thin-film sensors towards exposure of nitrogen dioxide. The lateral size of FG flakes was changed by additional stages of sonication and centrifugation. It was shown that ultrasonic treatment of fluorinated graphite resulted in partial defluorination of FG flakes for all fluorination degrees. The sensor properties of FG films were tested to 100 ppm NO_2_ at room and elevated temperatures. It was shown that the particle fraction affects the relative response via efficiency of charge transfer between the molecule and graphene layers. The sensor characteristics of FG enhanced with the fluorination loading. The fluorine groups act as a scattering center, dopant, and adsorption site, which improves electrical response and response/recovery rates. Regarding that, we stated a high efficiency of fluorinated graphites for the fabrication of thin-film graphene-based sensors due to their excellent exfoliation in organic solvents and tunable electronic properties.

## Figures and Tables

**Figure 1 materials-13-03538-f001:**
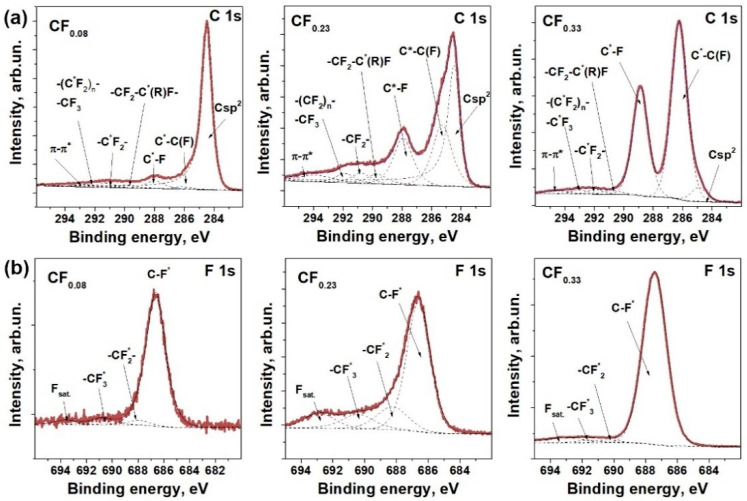
XPS C 1s (**a**) and F 1s (**b**) spectra of fluorinated graphites with the composition of layers of CF_0.08_, CF_0.23_, and CF_0.33_.

**Figure 2 materials-13-03538-f002:**
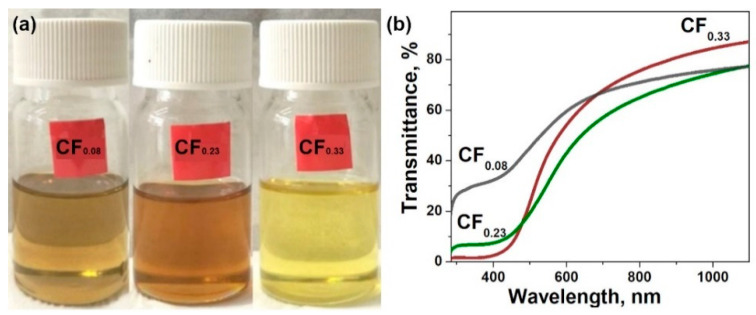
(**a**) Photographs of fluorinated graphene (FG) dispersions obtained from CF_0.08_, CF_0.23_, and CF_0.33_ (from left to right) samples; (**b**) optical absorption spectra in the visible and near-IR regions of FG dispersions in toluene.

**Figure 3 materials-13-03538-f003:**
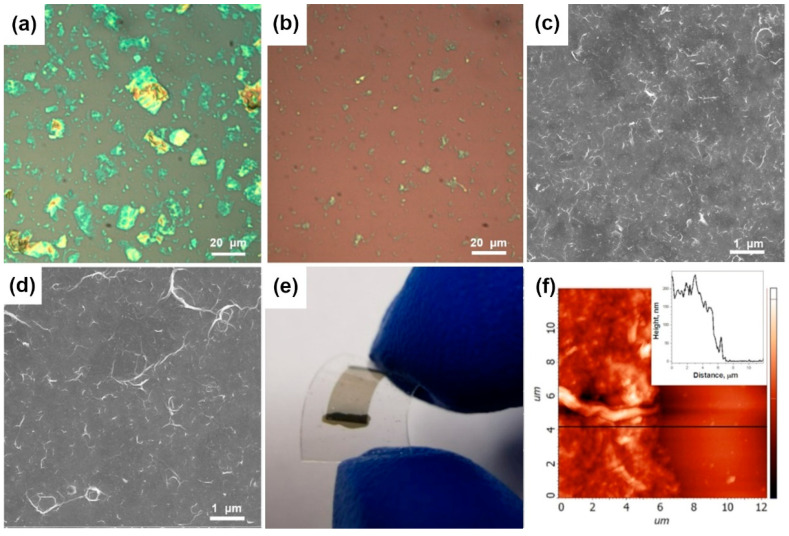
Optical images of FG particles after sedimentation (**a**) and centrifugation (**b**) steps. SEM images of FG films obtained from fluorinated graphites CF_0.08_ (**c**) and CF_0.23_ (**d**). Photo of FG film on PET substrate (**e**). AFM image of FG film (**f**), inset shows height profile along black line.

**Figure 4 materials-13-03538-f004:**
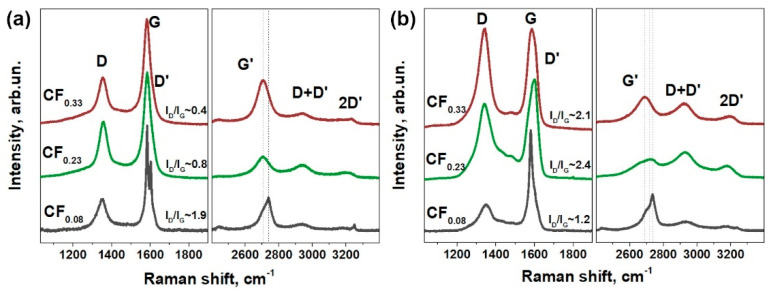
Raman spectra of fluorinated graphite powders (**a**) and FG films obtained from these powders (**b**). The spectra were normalized to the intensity of G peak.

**Figure 5 materials-13-03538-f005:**
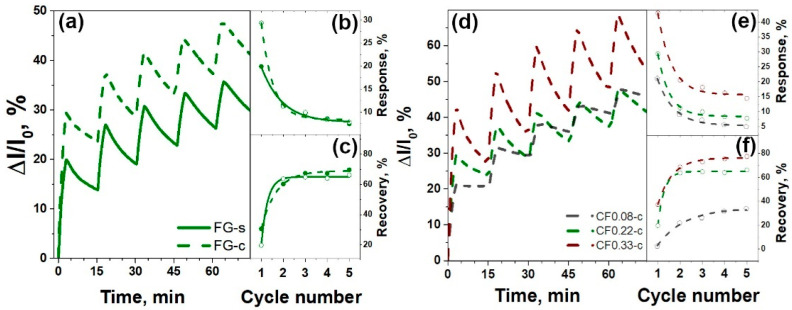
Run-to-run cycling of FG-s and FG-c sensors obtained from graphite fluoride CF_0.23_ (**a**) and FG-c sensors obtained from CF_0.08_, CF_0.23_, and CF_0.33_ samples (**d**) towards exposure of 100 ppm NO_2_ at room temperature. Relative response (**b,e**) and recovery (**c,f**) of the sensors vs. cycle number.

**Figure 6 materials-13-03538-f006:**
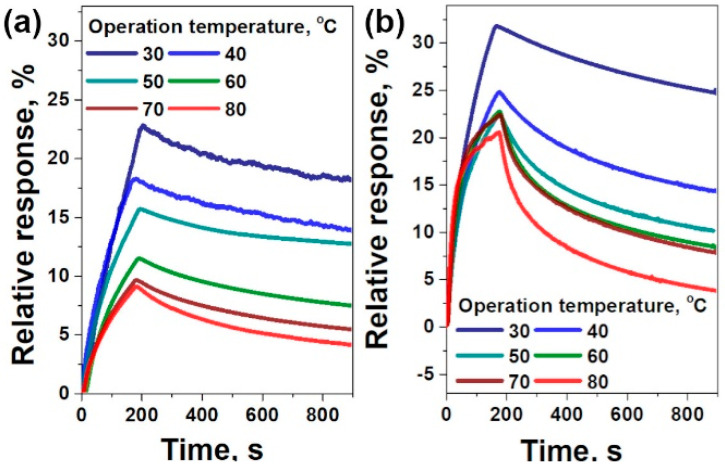
Sensor response of FG-c films prepared from graphite fluorides CF_0.08_ (**a**) and CF_0.33_ (**b**) towards 100 ppm NO_2_ at temperatures from 30 to 80 °C.

**Table 1 materials-13-03538-t001:** Sample name, fluorine content, and fraction of sp^2^-hybridized carbon regions in fluorinated graphite powders determined by XPS, thickness (t), sheet resistance (R_s_), resistivity (ρ) of the films and their sensor response and recovery.

Sample		F Content, at.%	C_sp2_, Fraction %	t, nm	R_s_, kΩ/□	ρ, Ω·m	Response, %	Recovery, %
CF_0.08_	FG-s	7	88	146	66.9	9.8 × 10^−2^	10	21
	FG-c			266	75.3	2.0 × 10^−2^	21	2
CF_0.23_	FG-s	19	69	210	163.9	3.4 × 10^−2^	15	30
	FG-c			210	120.9	2.5 × 10^−2^	29	19
CF_0.33_	FG-c	25	60	215	81960	17.6	43	37

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
