# Peer review of "Chemiresistive Properties of Imprinted Fluorinated Graphene Films"

_materials, 2020, doi:10.3390/ma13163538_

Round 1
Reviewer 1 Report
The work entitled “Chemrisistive properties of imprinted fluorinated graphene films” reported the relationship between the fluorination and sensor response. This paper can be published after modifying the following mentioned parts.
(1) Please check “chemrisistive” written in the title, it should be chemiresistive
(2) To convincedly show the response of centrifugation samples are higher than sediment samples, the comparison results of CF0.08-s and CF0.08-c, CF0.33-s and CF0.33-c are also needed.
(3) In the Results and Discussion part, the sample name in the paragraph and Figures is not consistent, as well as that in Table 1. It makes the readers confusing what are exactly the authors tested for the measurement. Please check it and use the same name to describe the same sample.
For example
① Page 5, the description is based on CF0.23-s, CF0.07-c, and CF0.33-c, but in the figure 4, the samples are CF0.22-s, CF0.22-c, CF0.07-c, and CF0.33-c. Also, in Table 1, the prepared samples are CF0.08, CF0.23, and CF0.33. It is quite confusing.
② Page 6, it appears CF0.05 and CF0.25 as well as CF0.07, they were not the samples shown in Table 1.
③ Page 7, the legend of Figure 5 is CF0.05 and CF0.2, but the figure content is CF0.07 and CF0.33.
(4) Page 5, line 171, delete which is, there is double which is in one sentence.
(5) Page 5, line 176, figure 3a should be figure 4a; and line 183, figure 3b should be figure 4b.
Author Response
Reviewer 1
We thank the Reviewer 1 for the examination of our manuscript and bellow give answers to each comment.
(1) Please check “chemrisistive” written in the title, it should be chemiresistive
Reply: We are very sorry for this misprint, the title of the manuscript was corrected.
(2) To convincedly show the response of centrifugation samples are higher than sediment samples, the comparison results of CF0.08-s and CF0.08-c, CF0.33-s and CF0.33-c are also needed.
Reply: The sensor responses of centrifugated and sediment sample have similar behavior for the FG films obtained from CF0.08 and CF0.023 materials. The film CF0.33-s is highly resistive (>200MOhm), so sensor measurement has not been obtained. The values of relative response and recovery of each samples were introduced in Table 1.
(3) In the Results and Discussion part, the sample name in the paragraph and Figures is not consistent, as well as that in Table 1. It makes the readers confusing what are exactly the authors tested for the measurement. Please check it and use the same name to describe the same sample. For example
① Page 5, the description is based on CF0.23-s, CF0.07-c, and CF0.33-c, but in the figure 4, the samples are CF0.22-s, CF0.22-c, CF0.07-c, and CF0.33-c. Also, in Table 1, the prepared samples are CF0.08, CF0.23, and CF0.33. It is quite confusing.
② Page 6, it appears CF0.05 and CF0.25 as well as CF0.07, they were not the samples shown in Table 1.
③ Page 7, the legend of Figure 5 is CF0.05 and CF0.2, but the figure content is CF0.07 and CF0.33.
Reply: We carefully checked the names of all samples and introduced the necessary corrections in the manuscript. The samples are named CF0.33, CF0.23, and CF0.08 according to the composition of the layers of parent fluorinated graphites determined from elemental analysis.
(4) Page 5, line 171, delete which is, there is double which is in one sentence.
Reply: The sentence was corrected.
(5) Page 5, line 176, figure 3a should be figure 4a; and line 183, figure 3b should be figure 4b.
Reply: New version of the manuscript has correct assignments to the figures.
Reviewer 2 Report
Sysoev et al. investigated the chemresistive features of fluorinated graphene flims. The research could have been more interesting after incorporating the following concerns:
1. 100 ppm is too high. please show lower ppm-level of detection like ref1 ref2 and cite them.
2. resistivity of both CF0.33 is huge. how it can respond better and faster?
3. could not find about CF0.66 (mentioned in abstract) in the main text
4. is there any possibility of nitrogen in the final synthesized structure since the main precursor contains nitrogen?
5. Raman peak around 2700 cm-1 is conventionally assigned as G'. 2D is conventionally used for two-dimensional and this peak is not related to D-peak (1350 cm-1). Please change and cite Ref
6. please maintain the same font in the Figure 4
7. there is a problem in assigning the Figure number (figure 4) in the text
8. please correct the references (ref no. 26, 27, 32,34)
9. In Figure 3, correct the captioning problem. Lebel the Raman spectra with sample. intensity should be expressed as arb. unit since a.u. is conventionally used for the astronomical unit
10. In table, please show the change in sp2-C content
Author Response
Reviewer 2
We thanks the Reviewer 2 for the comments, which are responded below.
Sysoev et al. investigated the chemresistive features of fluorinated graphene flims. The research could have been more interesting after incorporating the following concerns:
- 100 ppm is too high. please show lower ppm-level of detection like ref1 ref2 and cite them.
Reply: The aim of this work is to reveal the influence of composition and size of FG particles on the sensor properties of FG films. The concentration of NO2 equal to 100 ppm is chosen to be detectable for all samples. A lowering of the concentration produces too weak signal for particularly the FG sensor prepared from the CF0.08 fluorinated graphite.
- resistivity of both CF0.33 is huge. how it can respond better and faster?
Reply: There is no clear correlation between the resistivity and the response of a sample. The rate performance depends on the rate constant of adsorption of NO2 on the surface. In case of graphene material, it is determined by presence of defect, dopants or functional groups on the surface:
“Response and recovery of the FG sensor increased with the fluorine content (Table 1). Fluorinated graphene is a p-type semiconductor since fluorine is strong electron acceptor [22]. An increase of the content of covalently attached fluorine leads to an increase in the concentration of hole carriers in the material. Additionally, fluorination introduces adsorption sites near the fluorine groups [26, 48].”
- could not find about CF0.66 (mentioned in abstract) in the main text
Reply: All samples were correctly renamed in current version of the manuscript. The samples are named CF0.33, CF0.23, and CF0.08 according to the composition of the layers of parent fluorinated graphites determined from elemental analysis
- is there any possibility of nitrogen in the final synthesized structure since the main precursor contains nitrogen?
Reply: According to XPS the content of nitrogen in initial fluorinated graphites was ca. 1.3 at.% in the sample CF0.33 and ca. 0.9 at.% in the sample CF0.23. XPS was not able to detect nitrogen in CF0.88. Sonication of a fluorinated graphite sample in toluene and filtration of the obtained dispersion remove acetonitrile molecules, which were hosted between fluorinated graphite layers. If even some molecules remained in the films their content is too low to be datable by XPS and EDS.
- Raman peak around 2700 cm-1 is conventionally assigned as G'. 2D is conventionally used for two-dimensional and this peak is not related to D-peak (1350 cm-1). Please change and cite Ref
Reply: The interpretation of Raman spectra was corrected and additional references were provided: “Raman spectroscopy proved the functionalization of graphene layers in studied materials (Figure 4). Fluorination of graphite led to the appearance of defect-activated peaks D at 1351 cm–1 and D’ at 1604 cm–1 (Figure 4a), which correspond to the single-phonon intervalley and intravalley scattering processes, respectively. The intensity of these peaks relative to the G peak intensity changes non-monotonically with the fluorine content, which is similar to the previous results on functionalized graphenes [41]. The D’ peak is separated from G peak in the spectrum of CF0.08 powder and these peaks are merged when the fluorine content increases. All the single-phonon peaks are broaden for the FG films (Figure 4b) as compared to the powder parent materials. The intensity of D peak increases significantly due to formation of new edge states resulting from the exfoliation process. An increase of the fluorination degree leads to a blue shift of the two-phonon G’-peak for both powders and FG films, which evidences p-type doping of graphene [42].”
- please maintain the same font in the Figure 4
Reply: Figure 4 was changed and renamed to figure 5.
- there is a problem in assigning the Figure number (figure 4) in the text.
Reply: Sorry for the misprint related to the assignment of the figure, it was corrected.
- please correct the references (ref no. 26, 27, 32,34)
Reply: All references were сcorrected.
- In Figure 3, correct the captioning problem. Lebel the Raman spectra with sample. intensity should be expressed as arb. unit since a.u. is conventionally used for the astronomical unit
Reply: These remarks were taken into account in the new version of the manuscript.
- In table, please show the change in sp2-C content
Reply: The deconvolution of C 1s spectra of each sample were added to Figure 2. The sp2-C content is shown in the table 1.
Reviewer 3 Report
V.I. Sysoev et al. Prepared and characterized fluorinated graphene films. The manuscript is written clearly and concisely. The manuscript contains an original piece of work. The authors should make the major revisions before the manuscript can be accepted for publication in Materials:
1) x-ray photoelectron (XP) spectroscopy >> XPS is widely used
2) energy dispersion x-ray spectroscopy (EDS) >> Energy Dispersive X-ray Spectroscopy
3) Figure 3: XPS, show also the survey spectrum (not only C1s) and calculate C/F ratio.
4) Page 5: ED spectroscopy ??
5) Introduction should be extended about synthesis of halogenated graphenes. See eg. a) RSC Adv., 6, 2016, 66884–66892, b) Chem. Eur. J., 2016, 22, 17696-17703, c) Chem. Eur. J., 2017, 23, 10473–10479 and d) Nanoscale, 2015, 7, 13646-13655.
6) Why is there red colour in ref 25? And yellow in ref 13??
7) Pg. 2: Graphite fluoride were synthesized >> fluorides were / fluoride was
Author Response
Reviewer 3
We thank the Reviewer 3 for high evaluation of our work and below describe the changes made in the manuscript according to the remarks.
V.I. Sysoev et al. Prepared and characterized fluorinated graphene films. The manuscript is written clearly and concisely. The manuscript contains an original piece of work. The authors should make the major revisions before the manuscript can be accepted for publication in Materials:
1) x-ray photoelectron (XP) spectroscopy >> XPS is widely used
Reply: The required change was made in the text.
2) energy dispersion x-ray spectroscopy (EDS) >> Energy Dispersive X-ray Spectroscopy
Reply: We fixed this mistake.
3) Figure 3: XPS, show also the survey spectrum (not only C1s) and calculate C/F ratio.
Reply: The C/F ratios calculated from XPS survey spectra were introduced in the first paragraph of the part 3.1 Materials characterization: “The content of fluorine in the surface of the fluorinated graphites determined from the XPS survey spectra was ca. 7 at.% in CF0.08 sample, ca. 19 at.% in CF0.23 sample, and ca. 25 at.% in CF0.33 sample. Lower values given by XPS are due to partial hydrolysis of the sample surface by H2O present in laboratory air [32].” To characterize the C-F bonding in the fluorinated graphites we presented the XPS C 1s and F 1s spectra for all parent samples in Figure 1 and provided their description.
4) Page 5: ED spectroscopy ??
Reply: This misprint was corrected.
5) Introduction should be extended about synthesis of halogenated graphenes. See eg. a) RSC Adv., 6, 2016, 66884–66892, b) Chem. Eur. J., 2016, 22, 17696-17703, c) Chem. Eur. J., 2017, 23, 10473–10479 and d) Nanoscale, 2015, 7, 13646-13655.
Reply: We thank the Reviewer for this suggestion and introduced a couple of sentences on this issue in the Introduction part: “Halogenation is a promising way to obtained graphene derivatives possessing an energy gap, unique two-dimensional structure, and uniform distribution of functional groups on the surface [17]. As compared to bulky chlorine [18] and bromine [19] atoms, fluorine can attach to every carbon atom up to fully fluorinated graphene (FG) layer [20].”
6) Why is there red colour in ref 25? And yellow in ref 13??
Reply: Coloring of the references was deleted.
7) Pg. 2: Graphite fluoride were synthesized >> fluorides were / fluoride was
Reply: We fixed this misprint.
Reviewer 4 Report
Chemrisistive properties of imprinted fluorinated graphene films
I can recommend the paper for acceptance after the authors addressed the following issues and made corresponding corrections and additions in the manuscript.
- The logic of Novoselov and Geim's use of graphene for sensor applications was graphene’s high electron mobility. It is known that fluorination degrades mobility (although maybe induces the bandgap). What is the logic of using fluorinated graphene films for senor? Why not something else more conventional? Is it just a large surface-to-volume ratio of any 2D material? Some explanations would be useful. Do you know what is the mobility in your material and if there is a band gap?
- There have been a lot of literature on graphene sensors, which constitute the prior state of the art. These prior works should be cited and results compared when possible. The relevant papers are: S. Rumyantsev, et al., “Selective gas sensing with a single pristine graphene transistor,” Nano Lett., vol. 12, no. 5, pp. 2294–2298, 2012; IEEE Sens. J., vol. 13, no. 8, pp. 2818–2822, Aug. 2013; R. Samnakay, et al., Appl. Phys. Lett., vol. 106, no. 2, p. 23115, Jan. 2015.
- Figure 2 (a) and (b) do not look nice and do not provide much info. Are you sure you want to keep them in the main text?
- It would be good to compare the performance of these sensors with other 2D material sensors that do not have large mobility and have band gap. Example of relevant literature is G. Liu, et al., “Selective gas sensing with h-BN capped MoS2 heterostructure thin-film transistors,” IEEE Electron device Lett., vol. 36, no. 11, pp. 1202–1204, Nov. 2015.
Author Response
Reviewer 4
I can recommend the paper for acceptance after the authors addressed the following issues and made corresponding corrections and additions in the manuscript.
- The logic of Novoselov and Geim's use of graphene for sensor applications was graphene’s high electron mobility. It is known that fluorination degrades mobility (although maybe induces the bandgap). What is the logic of using fluorinated graphene films for senor? Why not something else more conventional? Is it just a large surface-to-volume ratio of any 2D material? Some explanations would be useful. Do you know what is the mobility in your material and if there is a band gap?
Reply: We explain motivation to use the fluorinated graphene as graphene-based sensor in the first paragraph of the Discussion part:
“Fluorination of graphene is especially suitable in that case because it introduces single-type functional groups at the basal plane and allows tuning the F/C ratio. The electrical transport properties of partially fluorinated graphene can be adjusted by tuning the fluorination degree for application in chemical sensors. The charge carrier mobility for single flake devices was found to increase with an increase of the fluorination degree, reaching 2000–3000 cm2V−1s−1 [48]. In case of thin films, the charge transports is largely affected by edge/edge, edge/plane and plane/plane junctions. Additionally, we have previously demonstrated that C–F groups modify the surface chemistry of graphene forming specific sites for molecule adsorption [31,48]. Similar effect has been achieved for plasma-fluorinated CVD-graphene, whose enhanced sensor performance at room temperature was attributed to p-type doped nature of FG and stronger physical adsorption of ammonia [26]. Park and co-authors have proposed to modify graphene oxide by fluorine and achieved better sensitivity of the fluorinated sensor to ammonia, however all obtained sensors failed to be recovered at room temperature [49]. ”
Comparing to graphene fluorinated graphene has semiconducting nature. Single flake devices based on low fluorinated graphene showed carrier mobility of few thousands of cm2V−1s−1. In case of thin films, the charge transport largely affected by flake-flake junctions, which decreases carrier mobility by 2-3 orders of magnitude. Despite of that, these values is high enough for low-frequency electronics such as chemical gas sensors, since the rate performance of sensor response limited by the adsorption processes, which has characteristic times from ~10 to ~1000 s.
The change in optical band gap depending on fluorine content in the studied samples is demonstrated in Figure 2b.
- There have been a lot of literature on graphene sensors, which constitute the prior state of the art. These prior works should be cited and results compared when possible. The relevant papers are: S. Rumyantsev, et al., “Selective gas sensing with a single pristine graphene transistor,” Nano Lett., vol. 12, no. 5, pp. 2294–2298, 2012; IEEE Sens. J., vol. 13, no. 8, pp. 2818–2822, Aug. 2013; R. Samnakay, et al., Appl. Lett., vol. 106, no. 2, p. 23115, Jan. 2015.
- Reply: We thank the Reviewer for the useful papers, which were cited in the Discussion part: “The fluctuations in the spectral density of the low-frequency current induced by some gases were proposed as the sensing parameter to enhance selectivity of graphene [44]. Surface functionalization and operation at elevated temperature reduce the response and recovery times of graphene-based sensors [45]. These sensing parameters can be controlled by density of the adsorption sites [46]. Fluorination of graphene is especially suitable in that case because it introduces single-type functional groups at the basal plane and allows tuning the F/C ratio.”
- Figure 2 (a) and (b) do not look nice and do not provide much info. Are you sure you want to keep them in the main text?
Reply: These figures are presented now in Figure 3. They demonstrate the difference in the particle size of FG after sedimentation and centrifugation steps that is important issue of our work.
- It would be good to compare the performance of these sensors with other 2D material sensors that do not have large mobility and have band gap. Example of relevant literature is G. Liu, et al., “Selective gas sensing with h-BN capped MoS2 heterostructure thin-film transistors,” IEEE Electron device Lett., vol. 36, no. 11, pp. 1202–1204, Nov. 2015.
Reply: According to Reviewer’s requirement, we added a few sentences at the end of the Discussion part: “Other two-dimensional materials, particularly MoS2, WS2, black phosphorus, are widely examined for gas sensing due to their high surface-to-volume ratio and tunable band gap [55]. Unlike the FG, the fabrication of sensors from the above materials is currently costly. Contaminations and easy oxidation of surfaces require the capping step to prevent problems with reliability and stability of these devices [56].”
Round 2
Reviewer 2 Report
I am quite satisfied with the updated manuscript and hence recommending to accept for publication
Reviewer 3 Report
The authors improved the manuscript significantly. Now, it can be accepted for publication in Materials.